# DBLP: Noise Bridge Consistency Distillation For Efficient And Reliable Adversarial Purification

## Abstract

Recent advances in deep neural networks (DNNs) have led to remarkable success across a wide range of tasks. However, their susceptibility to adversarial perturbations remains a critical vulnerability. Existing diffusion-based adversarial purification methods often require intensive iterative denoising, severely limiting their practical deployment. In this paper, we propose Diffusion Bridge Distillation for Purification (DBLP), a novel and efficient diffusion-based framework for adversarial purification. Central to our approach is a new objective, noise bridge distillation, which constructs a principled alignment between the adversarial noise distribution and the clean data distribution within a latent consistency model (LCM). To further enhance semantic fidelity, we introduce adaptive semantic enhancement, which fuses multi-scale pyramid edge maps as conditioning input to guide the purification process. Extensive experiments across multiple datasets demonstrate that DBLP achieves state-of-the-art (SOTA) robust accuracy, superior image quality, and around 0.2s inference time, marking a significant step toward real-time adversarial purification.

## 1 Introduction

Deep neural networks (DNNs) have achieved remarkable success across a wide range of tasks in recent years. However, their widespread deployment has raised increasing concerns about their security and robustness He et al. (2016); Liu et al. (2021). It is now well-established that DNNs are highly vulnerable to adversarial attacks Szegedy et al. (2014a), wherein imperceptible, carefully crafted perturbations are added to clean inputs to generate adversarial examples that can mislead the model into producing incorrect outputs Huang & Shen (2025).

To address this issue, adversarial training (AT) Madry et al. (2019) has been proposed, which retrains classifiers using adversarial examples. However, AT suffers from high computational cost and poor generalization to unseen threats, limiting its applicability in real-world adversarial defense scenarios.

In contrast, adversarial purification (AP) has emerged as a compelling alternative due to its stronger generalization capabilities, and its plug-and-play nature, requiring no classifier retraining. AP methods utilize generative models as a preprocessing step to transform adversarial examples into purified ones, which are then fed into the classifier. The recent advances in diffusion models Ho et al. (2020) have further propelled the development of AP. These models learn to transform simple distributions into complex data distributions through a forward noising and reverse denoising process. Crucially, this iterative denoising mechanism aligns well with the goal of removing adversarial perturbations, making diffusion models a natural fit for AP tasks Nie et al. (2022).

However, existing diffusion-based purification approaches suffer from a critical limitation: they require multiple iterative denoising steps, resulting in prohibitively slow inference, which severely restricts their use in latency-sensitive applications such as autonomous driving Chi et al. (2024) and industrial manufacturing Wang et al. (2025). Moreover, most of these methods rely on a key assumption that the distributions of clean and adversarial samples converge after a certain number of forward diffusion steps. This allows the use of pretrained diffusion models, originally designed for generative tasks, to purify adversarial samples. However, this assumption only holds when the diffusion time horizon is sufficiently large. Empirical evidence from DiffPure Nie et al. (2022)

suggests that excessive diffusion steps can lead to significant loss of semantic content, rendering accurate reconstruction of clean images infeasible.

In this paper, we propose Diffusion Bridge Distillation for Purification (DBLP), a novel framework designed to simultaneously address the two key limitations of existing diffusion-based adversarial purification methods: low inference efficiency and detail degradation. At its core, DBLP introduces a noise-bridged alignment strategy within the Latent Consistency Model Luo et al. (2023a), effectively bridging adversarial noise and clean targets during the consistency distillation process to better align with the purification objective. By leveraging noise bridge distillation, DBLP enables direct recovery of clean samples from diffused adversarial inputs using an ODE solver. To further mitigate detail loss caused by fewer denoising steps, we introduce adaptive semantic enhancement, a lightweight yet effective conditioning mechanism that utilizes multi-scale pyramid edge maps to capture fine-grained structural features. These semantic priors are injected into inference to enhance content preservation. DBLP achieves SOTA robust accuracy across multiple benchmark datasets while substantially reducing inference latency, requiring only 0.2 seconds per sample, thus making real-time adversarial purification feasible without compromising visual quality.

In summary, our contributions can be summarized as follows:

- We propose DBLP, a novel diffusion-based adversarial purification framework that significantly accelerates inference while improving purification performance and visual quality.
- We introduce a noise bridge distillation objective tailored for adversarial purification within the latent consistency model, effectively setting a bridge between adversarial noise and clean samples. Additionally, we design an adaptive semantic enhancement module that improves the model's ability to retain fine-grained image details during purification.
- Comprehensive experiments across multiple benchmark datasets demonstrate that our method achieves SOTA performance in terms of robust accuracy, inference efficiency, and image quality, moving the field closer to practical real-time adversarial purification systems.

## 2 RELATED WORK

### 2.1 ADVERSARIAL TRAINING

Adversarial training is a prominent defense strategy against adversarial attacks Goodfellow et al. (2015), which enhances model robustness by retraining the model on perturbed adversarial examples Lau et al. (2023). A substantial body of research has demonstrated its efficacy in adversarial defense. Notable methods include min-max optimization framework Madry et al. (2018), TRADES which balances robustness and accuracy via a regularized loss Zhang et al. (2019), and techniques like local linearization Qin et al. (2019) and mutual information optimization Zhou et al. (2022). Despite its strong robustness, adversarial training suffers from several notable drawbacks. It often generalizes poorly to unseen attacks Laidlaw et al. (2021), and it incurs significant computational overhead due to the necessity of retraining the entire model. Moreover, it typically leads to a degradation in clean accuracy Wong et al. (2020).

### 2.2 ADVERSARIAL PURIFICATION

Adversarial purification represents an alternative and effective defense strategy against adversarial attacks that circumvents the need for retraining the model. The core idea is to employ generative models to pre-process adversarially perturbed images, yielding purified versions that are subsequently fed into the classifier. Early efforts in this domain leveraged GANs Samangouei et al. (2018) or score-based matching techniques Yoon et al. (2021); Song et al. (2021) to successfully restore adversarial images. DiffPure Nie et al. (2022) advanced this with diffusion models, inspiring follow-ups like adversarially guided denoising Wang et al. (2022); Wu et al. (2022), improved evaluation frameworks Lee & Kim (2023), gradient-based purification Zhang et al. (2023a), dual-phase guidance Song et al. (2024), and adversarial diffusion bridges Li et al. (2025). Despite their promising results, these methods exhibit certain limitations. Many approaches rely on auxiliary classifiers, which often compromise generalization performance. Others involve iterative inference procedures that are computationally intensive and time-consuming, thereby limiting their practicality in real-time or resource-constrained scenarios.

## 2.3 DIFFUSION MODELS

Diffusion models Ho et al. (2020), originally introduced to enhance image generation capabilities, have since demonstrated remarkable success across various domains, including video synthesis Ho et al. (2022) and 3D content generation Luo & Hu (2021). As a class of score-based generative models, diffusion models operate by progressively corrupting images with Gaussian noise in the forward process, and subsequently generating samples by denoising in the reverse process Huang & Tang (2025). Given a pre-defined forward trajectory $\{\mathbf{x}_t\}_{t \in [0,T]}$, indexed by a continuous time variable $t$, the forward process can be effectively modeled using a widely adopted stochastic differential equation (SDE) Karras et al. (2022):

$$\mathrm{d}\mathbf{x}_t = \boldsymbol{\mu}(\mathbf{x}_t, t)\mathrm{d}t + \sigma(t)\mathrm{d}\mathbf{w}_t, \tag{1}$$

where $\boldsymbol{\mu}(\mathbf{x}_t, t)$ and $\sigma(t)$ denote the drift and diffusion coefficients, respectively, while $\{\mathbf{w}_t\}_{t \in [0,T]}$ represents a standard $d$-dimensional Brownian motion. Let $p_t(\mathbf{x})$ denote the marginal distribution of $\mathbf{x}_t$ at time $t$, and $p_{\mathrm{data}}(\mathbf{x})$ represent the distribution of the original data, then $p_0(\mathbf{x}) = p_{\mathrm{data}}(\mathbf{x})$.

Remarkably, Song et al. (2021) established the existence of an ordinary differential equation (ODE), referred to as the *Probability Flow* (PF) ODE, whose solution trajectories share the same marginal probability densities $p_t(\mathbf{x})$ as those of the forward SDE:

$$\mathrm{d}\mathbf{x}_t = \left[ \boldsymbol{\mu}\left(\mathbf{x}_t, t\right) - \frac{1}{2}\sigma(t)^2 \nabla \log p_t(\mathbf{x}_t) \right] \mathrm{d}t. \tag{2}$$

For sampling, a score model $s_\phi(\mathbf{x}, t) \approx \nabla \log p_t(\mathbf{x})$ is first trained via score matching to approximate the gradient of the log-density at each time step. This learned score function is then substituted into Equation equation 2 to obtain an empirical estimate of the PF ODE:

$$\frac{\mathrm{d}\mathbf{x}_t}{\mathrm{d}t} = \boldsymbol{\mu}\left(\mathbf{x}_t, t\right) - \frac{1}{2}\sigma(t)^2 s_\phi(\mathbf{x}_t, t). \tag{3}$$

# 3 PRELIMINARIES

## 3.1 PROBLEM FORMULATION

Adversarial attacks were first introduced by Szegedy et al. (2014b), who revealed the inherent vulnerability of neural networks to carefully crafted perturbations. An adversarial example $\mathbf{x}_{\mathrm{adv}}$ is visually and numerically close to a clean input $\mathbf{x}$, yet it is deliberately designed to mislead a classifier C into assigning it to an incorrect label, rather than the true class $y_{\mathrm{true}}$, formally expressed as:

$$\arg\max_y \mathrm{C}(y|\mathbf{x}_{\mathrm{adv}}) \neq y_{\mathrm{true}}, \tag{4}$$

with the constraint of $\|\mathbf{x}_{\mathrm{adv}} - \mathbf{x}\| \leq \epsilon$, where $\epsilon$ is the perturbation threshold.

The concept of adversarial purification is to transform the adversarial input $\mathbf{x}_{\mathrm{adv}}$ into a purified sample $\mathbf{x}_{\mathrm{pur}}$ before passing it to the classifier C, such that $\mathbf{x}_{\mathrm{pur}}$ closely approximates the clean sample $\mathbf{x}$ and yields the correct classification outcome. This process can be formulated as:

$$\max_{\mathrm{P}} \mathrm{C}(y_{\mathrm{true}}|\mathrm{P}(\mathbf{x}_{\mathrm{adv}})), \tag{5}$$

where $\mathrm{P} : \mathbb{R}^d \to \mathbb{R}^d$ is the purification function.

## 3.2 CONSISTENCY MODELS

The long inference time of diffusion models is a well-known limitation, prompting the introduction of the Consistency Model Song et al. (2023), which enables the sampling process to be reduced to just a few steps, or even a single step. It proposes learning a direct mapping from any point $\mathbf{x}_t$ along the PF ODE trajectory $\{\mathbf{x}_t\}_{t \in [0,T]}$ back to its starting point, referred to as the consistency function, denoted as $\boldsymbol{f} : (\mathbf{x}_t, t) \mapsto \mathbf{x}_\epsilon$, where $\mathbf{x}_\epsilon$ represents the starting state at a predefined small positive value $\epsilon$. The *self-consistency* property of this function can be formalized as:

$$\boldsymbol{f}(\mathbf{x}_t, t) = \boldsymbol{f}(\mathbf{x}_{t'}, t') \quad \forall t, t' \in [0, T]. \tag{6}$$

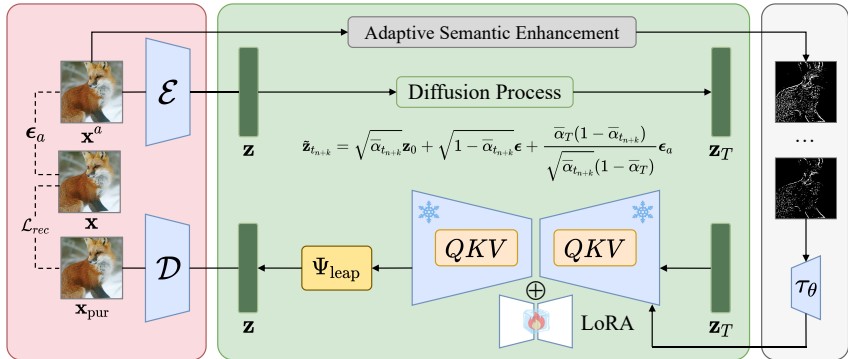

Figure 1: The overview structure of our DBLP. An adversary perturbs a clean image $\mathbf{x}$ with noise $\boldsymbol{\epsilon}_a$ into an adversarial example $\mathbf{x}^a$, we first encode it into the latent space using the encoder $\mathcal{E}$ to obtain the latent representation $\mathbf{z}$, followed by noise injection as defined in Equation equation 11. During training, we adopt a modified LCM-LoRA framework to perform noise bridge consistency distillation on the diffusion model, and employ a leapfrog ODE solver to accelerate sampling. During inference, we introduce adaptive semantic enhancement, using the weighted fusion of pyramid edge maps as a semantic-preserving condition to guide the purification process. The final purified image $\mathbf{x}_{\text{pur}}$ is then recovered via the decoder $\mathcal{D}$.

The goal of the consistency model $\boldsymbol{f}_{\boldsymbol{\theta}}$ is to estimate the underlying consistency function $\boldsymbol{f}$ by enforcing the *self-consistency* property. The model $\boldsymbol{f}_{\boldsymbol{\theta}}$ can be parameterized as:

$$\boldsymbol{f}_{\boldsymbol{\theta}}(\mathbf{x}, t) = c_{\text{skip}}(t)\mathbf{x} + c_{\text{out}}(t)F_{\boldsymbol{\theta}}(\mathbf{x}, t), \tag{7}$$

where $c_{\text{skip}}(t)$ and $c_{\text{out}}(t)$ are differentiable functions. To satisfy the boundary condition $\boldsymbol{f}(\mathbf{x}_\epsilon, \epsilon) = \mathbf{x}_\epsilon$, we have $c_{\text{skip}}(\epsilon) = 1$ and $c_{\text{out}}(\epsilon) = 0$.

Building on this, the Latent Consistency Model Luo et al. (2023a) extends the consistency model to the latent space using an auto-encoder Rombach et al. (2022). In this setting, the consistency function conditioned on $\mathbf{c}$ is defined as $\boldsymbol{f}_{\boldsymbol{\theta}} : (\mathbf{z}_t, \mathbf{c}, t) \mapsto \mathbf{z}_\epsilon$. To fully leverage the capabilities of a pretrained text-to-image model, LCM parameterizes the consistency model as:

$$\boldsymbol{f}_{\boldsymbol{\theta}}(\mathbf{z}, \mathbf{c}, t) = c_{\text{skip}}(t)\mathbf{z} + c_{\text{out}}(t)\left(\frac{\mathbf{z} - \sigma_t\hat{\boldsymbol{\epsilon}}_\theta(\mathbf{z}, \mathbf{c}, t)}{\alpha_t}\right), \tag{8}$$

LCM-LoRA Luo et al. (2023b) proposes distilling LCM using LoRA, significantly reducing the number of trainable parameters and thereby greatly decreasing training time and computational cost.

## 4 METHODOLOGY

### 4.1 OVERALL FRAMEWORK

In this work, we aim to accelerate the purification backbone using a consistency distillation-inspired approach. Noting that the starting and ending points of the ODE trajectory respectively contain and exclude adversarial perturbations, we propose Noise Bridge Distillation in Section 4.2 to explicitly align the purification objective.

To achieve acceleration, we leverage the Latent Consistency Model with LoRA-based distillation and introduce a leapfrog ODE solver for efficient sampling. During inference, as detailed in Section 4.3, we propose Adaptive Semantic Enhancement, which fuses pyramid edge maps into a semantic-preserving condition to guide the diffusion model toward effective purification.

### 4.2 NOISE BRIDGE DISTILLATION

Following LCM Luo et al. (2023a), let $\mathcal{E}$ and $\mathcal{D}$ denote the encoder and decoder that map images to and from the latent space, respectively. Given an image $\mathbf{x}$, its latent representation is $\mathbf{z} = \mathcal{E}(\mathbf{x})$.

---

**Algorithm 1** Noise Bridge Distillation

---

**Input:** Dataset $\mathcal{D}$, LCM $\boldsymbol{f_\theta}$ and its initial model parameter $\boldsymbol{\theta}$, classifier C, ground truth label $y_{\text{true}}$, Leapfrog ODE solver $\Psi_{\text{leap}}$, distance metric $d(\cdot, \cdot)$, EMA rate $\mu$, noise schedule $\alpha_t$, skip interval $k$, encoder $\mathcal{E}$;
   $\quad \boldsymbol{\theta}^- \leftarrow \boldsymbol{\theta}$;
 1: **while** not convergence **do**
 2: $\quad$ Sample $\mathbf{x} \sim \mathcal{D}, n \sim \mathcal{U}\left[1, N-k\right]$;
 3: $\quad \mathbf{z} = \mathcal{E}\left(\mathbf{x}\right)$;
 4: $\quad \boldsymbol{\epsilon}_a = \arg\max_\epsilon \mathcal{L}(\text{C}(\mathcal{D}(\mathbf{z} + \boldsymbol{\epsilon})), y_{\text{true}})$;
 5: $\quad \tilde{\mathbf{z}}_{t_{n+k}} = \sqrt{\alpha_{t_{n+k}}}\mathbf{z}_0 + \sqrt{1 - \bar{\alpha}_{t_{n+k}}}\boldsymbol{\epsilon} + \frac{\bar{\alpha}_T(1 - \bar{\alpha}_{t_{n+k}})}{\sqrt{\bar{\alpha}_{t_{n+k}}}(1 - \bar{\alpha}_T)}\boldsymbol{\epsilon}_a$;
 6: $\quad \hat{\mathbf{z}}_{t_n}^{\Psi_{\text{leap}}} \leftarrow \mathbf{z}_{t_{n+k}} + \Psi(\mathbf{z}_{t_{n+k}}, t_{n+k}, t_n, \varnothing)$;
 7: $\quad \mathcal{L}_{\text{CD}}\left(\boldsymbol{\theta}, \boldsymbol{\theta}^-\right) = d\left(\boldsymbol{f_\theta}\left(\tilde{\mathbf{z}}_{t_{n+k}}, \varnothing, t_{n+k}\right), \boldsymbol{f}_{\boldsymbol{\theta}^-}\left(\hat{\mathbf{z}}_{t_n}^{\Psi_{\text{leap}}}, \varnothing, t_n\right)\right)$;
 8: $\quad \mathcal{L}_{\text{rec}}\left(\boldsymbol{\theta}\right) = d\left(\boldsymbol{f_\theta}\left(\tilde{\mathbf{z}}_t, \varnothing, t\right), \mathbf{z}\right)$;
 9: $\quad \mathcal{L}\left(\boldsymbol{\theta}, \boldsymbol{\theta}^-\right) = \mathcal{L}_{\text{CD}}\left(\boldsymbol{\theta}, \boldsymbol{\theta}^-\right) + \lambda_{\text{rec}}\mathcal{L}_{\text{rec}}\left(\boldsymbol{\theta}\right)$;
10: $\quad \boldsymbol{\theta} \leftarrow \boldsymbol{\theta} - \eta\nabla_\theta\mathcal{L}(\boldsymbol{\theta}, \boldsymbol{\theta}^-)$;
11: $\quad \boldsymbol{\theta}^- \leftarrow \text{sg}(\mu\boldsymbol{\theta}^- + (1-\mu)\boldsymbol{\theta})$
12: **end while**

---

Unlike DDPM, DBLP includes adversarial perturbations $\boldsymbol{\epsilon}_a$ at the start of the noising process, such that $\mathbf{z}_0^a = \mathbf{z}_0 + \boldsymbol{\epsilon}_a$. The forward process is then $\mathbf{x}_t^a = \sqrt{\bar{\alpha}_t}\mathbf{x}_0^a + \sqrt{1 - \bar{\alpha}_t}\boldsymbol{\epsilon}$, where $\boldsymbol{\epsilon} \sim \mathcal{N}(\mathbf{0}, \mathbf{I})$.

Our objective is to learn a trajectory that maps the diffused adversarial distribution ($\mathbf{z}_T^a$) back to the clean data distribution ($\mathbf{z}_0$). Notably, the starting point of this trajectory contains adversarial noise, whereas the endpoint does not. Therefore, we aim to find a consistency model $\boldsymbol{f_\theta}$ that satisfies: $\boldsymbol{f_\theta}(\mathbf{z}_t^a, \varnothing, t) = \boldsymbol{f_\theta}(\mathbf{z}_t, \varnothing, t) = \mathbf{z}_\epsilon$, where $\mathbf{z}_\epsilon \approx \mathbf{z}_0$ denotes the limiting state of $\mathbf{z}_t$ as $t \to 0$. However, this contradicts Equation equation 7, as the trajectories initiated from $\mathbf{z}_t$ and $\mathbf{z}_t^a$ are misaligned, causing $\boldsymbol{f_\theta}(\mathbf{z}_t^a, \varnothing, t) - \boldsymbol{f_\theta}(\mathbf{z}_t, \varnothing, t) \to \boldsymbol{\epsilon}_a$. To explicitly reconcile this discrepancy, we introduce a coefficient $k_t$ and define an adjusted latent variable $\tilde{\mathbf{x}}_t$ to align the trajectories accordingly:

$$\tilde{\mathbf{z}}_t = \mathbf{z}_t^a - k_t\boldsymbol{\epsilon}_a, \tag{9}$$

with $k_0 = 1$ and $k_T = 0$. Our goal is to ensure that the sampling distribution during the denoising process is independent of the adversarial perturbation $\boldsymbol{\epsilon}_a$. Although $\boldsymbol{\epsilon}_a$ can be computed during training as $\boldsymbol{\epsilon}_a = \arg\max_\epsilon \mathcal{L}(\text{C}(\mathcal{D}(\mathbf{z} + \boldsymbol{\epsilon})), y_{\text{true}})$, its exact value is unknown at inference time. Leveraging Bayes' theorem and the properties of Gaussian distributions, we achieve this by selecting the value of coefficient $k_t$ such that the term involving $\boldsymbol{\epsilon}_a$ is eliminated. After a series of derivations, we obtain an explicit closed-form expression for $k_t$:

$$k_t = \sqrt{\bar{\alpha}_t} - \frac{\bar{\alpha}_T(1 - \bar{\alpha}_t)}{\sqrt{\bar{\alpha}_t}(1 - \bar{\alpha}_T)}, 0 \le t \le T, \tag{10}$$

which satisfies $k_0 = 1$ and $k_T = 0$. Thus the $\tilde{\mathbf{z}}_t$ is constructed as:

$$\tilde{\mathbf{z}}_t = \sqrt{\bar{\alpha}_t}\mathbf{z}_0 + \sqrt{1 - \bar{\alpha}_t}\boldsymbol{\epsilon} + \frac{\bar{\alpha}_T(1 - \bar{\alpha}_t)}{\sqrt{\bar{\alpha}_t}(1 - \bar{\alpha}_T)}\boldsymbol{\epsilon}_a, \tag{11}$$

In this way, the sampling process doesn't require $\boldsymbol{\epsilon}_a$. The full proof is provided in Appendix A.2.

Accordingly, based on the loss function introduced in LCM Luo et al. (2023a), our consistency distillation loss can be formulated as:

$$\mathcal{L}_{CD}\left(\boldsymbol{\theta}, \boldsymbol{\theta}^-\right) = \mathbb{E}_{\mathbf{z}, n}\left[d\left(\boldsymbol{f_\theta}\left(\tilde{\mathbf{z}}_{t_{n+k}}, \varnothing, t_{n+k}\right), \boldsymbol{f}_{\boldsymbol{\theta}^-}\left(\hat{\mathbf{z}}_{t_n}^\Psi, \varnothing, t_n\right)\right)\right], \tag{12}$$

where $d(\cdot, \cdot)$ denotes a distance metric, and $\Psi(\cdot, \cdot, \cdot, \cdot)$ represents the DDIM Song et al. (2022) PF ODE solver $\Psi_{\text{DDIM}}$. The term $\hat{\mathbf{z}}_{t_n}^\Psi$ refers to the solution estimated by the solver when integrating from $t_{n+k}$ to $t_n$:

$$\hat{\mathbf{z}}_{t_n}^\Psi \leftarrow \mathbf{z}_{t_{n+k}} + \Psi(\mathbf{z}_{t_{n+k}}, t_{n+k}, t_n, \varnothing). \tag{13}$$

Following Kim et al. (2024), we also incorporate a reconstruction-like loss that leverages clean images to better align the distillation training process with the purification objective:

$$\mathcal{L}_{\text{rec}}\left(\boldsymbol{\theta}\right) = d\left(\boldsymbol{f_\theta}\left(\tilde{\mathbf{z}}_t, \varnothing, t\right), \mathbf{z}\right) \tag{14}$$

The training algorithm is detailed in Alg. 1.

Table 1: Clean Accuracy and Robust Accuracy (%) results on CIFAR-10. Avg. denotes the average robust accuracy across three types of attack threats, vanilla refers to models without any adversarial defense mechanism. The best results are **bolded**, and the second best results are underlined.

| Architecture | Type | Method | Clean Acc. | Robuse Acc. | | | |
|---|---|---|---|---|---|---|---|
| | | | | $\ell_\infty$ | $\ell_1$ | $\ell_2$ | Avg. |
| WRN-70-16 | − | Vanilla | 96.36 | 0.00 | 0.00 | 0.00 | 0.00 |
| WRN-70-16 | AT | Gowal et al. (2021a) | 91.10 | 65.92 | 8.26 | 27.56 | 33.91 |
| WRN-70-16 | | Rebuffi et al. (2021) | 88.54 | 64.26 | 12.06 | 32.29 | 36.20 |
| WRN-70-16 | | Aug. w/ Diff Gowal et al. (2021b) | 88.74 | 66.18 | 9.76 | 28.73 | 34.89 |
| WRN-70-16 | | Aug. w/ Diff Wang et al. (2023) | 93.25 | 70.72 | 8.48 | 28.98 | 36.06 |
| MLP+WRN-28-10 | AP | Shi et al. (2021) | 91.89 | 4.56 | 8.68 | 7.25 | 6.83 |
| UNet+WRN-70-16 | | Yoon et al. (2021) | 87.93 | 37.65 | 36.87 | 57.81 | 44.11 |
| UNet+WRN-70-16 | | GDMP Wang et al. (2022) | 93.16 | 22.07 | 28.71 | 35.74 | 28.84 |
| UNet+WRN-70-16 | | ScoreOpt Zhang et al. (2023a) | 91.41 | 13.28 | 10.94 | 28.91 | 17.71 |
| UNet+WRN-70-16 | | Purify++ Zhang et al. (2023b) | 92.18 | 43.75 | 39.84 | 55.47 | 46.35 |
| UNet+WRN-70-16 | | DiffPure Nie et al. (2022) | 92.50 | 42.20 | 44.30 | 60.80 | 49.10 |
| UNet+WRN-70-16 | | ADBM Li et al. (2025) | 91.90 | 47.70 | 49.60 | 63.30 | 53.50 |
| UNet+WRN-70-16 | | DBLP (Ours) | **94.8** | **58.4** | **64.4** | **59.4** | **60.73** |

**Leapfrog Solver**   To enhance the dynamical interpretability of the sampling process, we refine the DDIM-based PF ODE solver using a leapfrog-inspired mechanism. Specifically, we decompose the prediction into a position-like estimate of the clean image and a velocity-like estimate of the noise, which are then updated jointly through a first-order leapfrog integration step Verlet (1967):

$$\mathbf{z}_{t-1} = \mathbf{z}_0 + h \cdot \mathbf{v}_{1/2}, \tag{15}$$

where $\mathbf{z}_t = \sqrt{\bar{\alpha}_{t-1}} \cdot \hat{\mathbf{z}}_0$ and $\mathbf{v}_0 = \sqrt{1 - \bar{\alpha}_{t-1}} \cdot \hat{\boldsymbol{\epsilon}}$, while $\mathbf{v}_{1/2} = 2\mathbf{v}_0$ serves as the midpoint velocity estimate.

### 4.3 Adaptive Semantic Enhanced Purification

Although diffusion models are effective at learning the denoising process from noise to images, relying solely on this process often leads to the loss of fine-grained details Berrada et al. (2025). While OSCP Lei et al. (2025) attempts to mitigate this by incorporating edge maps to enhance structural information, it uses fixed-threshold Canny edge detection Canny (1986), which lacks adaptability to varying attack intensities. Moreover, adversarial perturbations introduce noise that can interfere with accurate edge extraction. To address these issues, we propose Adaptive Semantic Enhancement, a non-trainable, computationally efficient module to aggregate multi-scale edge information, enhancing structural integrity and detail preservation.

Given an adversarial image $\mathbf{x}_0^a \in \mathbb{R}^{H \times W \times 3}$, we construct an $L$-level Gaussian blur pyramid and apply adaptive thresholding at each level $l$ to compute the corresponding edge map:

$$\mathbf{E}_l = \text{Canny}(\text{GaussianBlur}(\mathbf{x}_0^a, \sigma_l)), \tag{16}$$

where the thresholds are calculated using Otsu Otsu (1979) algorithm.

We employ a gradient-guided mechanism to fuse edge maps across different scales. We first upsample all edge maps to a unified resolution $\mathbf{E}_l$, then use gradient consistency to compute the weights for each scale:

$$\mathbf{A}_l = \frac{\exp\left(-\|\nabla \mathbf{x}_0^a - \nabla \tilde{\mathbf{E}}_l\|_2 / T^*\right)}{\sum_{k=1}^{L} \exp\left(-\|\nabla \mathbf{x}_0^a - \nabla \tilde{\mathbf{E}}_k\|_2 / T^*\right)} \tag{17}$$

where $T^*$ is the temperature parameter. Finally the fused edge map is:

$$\mathbf{E}_{\text{fused}} = \sum_{l=1}^{L} \mathbf{A}_l \odot \tilde{\mathbf{E}}_l \tag{18}$$

We then use $\mathbf{E}_{\text{fused}}$ as a condition in the LCM, resulting in a semantically enhanced purified image.

Table 2: Clean Accuracy and Robust Accuracy (%) results on ImageNet. The default setting for attack is $\epsilon = 4/255$. The best results are **bolded**, and the second best results are underlined.

| Method | Type | Attack | Standard Acc. | Robust Acc. | Architecture |
|---|---|---|---|---|---|
| w/o Defense | – | PGD-100 | 80.55 | 0.01 | Res-50 |
| Schott et al. (2019) | AT | PGD-40 | 72.70 | 47.00 | Res-152 |
| Wang et al. (2020) | | PGD-100 | 53.83 | 28.04 | Res-50 |
| ConvStem Singh et al. (2023) | | AutoAttack | 77.00 | 57.70 | ConvNeXt-L |
| MeanSparse Amini et al. (2024) | | AutoAttack | 77.96 | 59.64 | ConvNeXt-L |
| DiffPure Nie et al. (2022) | AP | PGD-100 | 68.22 | 42.88 | Res-50 |
| DiffPure Nie et al. (2022) | | AutoAttack | 71.16 | 44.39 | WRN-50-2 |
| Bai et al. (2024) | | PGD-200 ($\epsilon = 8/255$) | 70.41 | 41.70 | Res-50 |
| Lee & Kim (2023) | | PGD+EOT | 70.74 | 42.15 | Res-50 |
| Lin et al. (2025) | | PGD+EOT | 68.75 | 45.90 | Res-50 |
| Zollicoffer et al. (2025) | | PGD-200 ($\epsilon = 8/255$) | 73.98 | 56.54 | Res-50 |
| MimicDiffusion Song et al. (2024) | | AutoAttack | 66.92 | 61.53 | Res-50 |
| ScoreOpt Zhang et al. (2023a) | | Transfer-PGD | 71.68 | 62.10 | WRN-50-2 |
| Pei et al. (2025) | | PGD-200 ($\epsilon = 8/255$) | 77.15 | 65.04 | Res-50 |
| OSCP Lei et al. (2025) | | PGD-100 | 77.63 | 73.89 | Res-50 |
| DBLP (Ours) | | PGD-100 | **78.2** | **75.6** | Res-50 |
| DBLP (Ours) | | AutoAttack | 78.0 | 74.8 | Res-50 |
| DBLP (Ours) | | PGD-200 ($\epsilon = 8/255$) | 77.4 | 74.2 | Res-50 |

# 5 EXPERIMENTS

## 5.1 EXPERIMENTAL SETTINGS

**Datasets** We conduct extensive experiments to validate the effectiveness and efficiency of our proposed method across several widely-used datasets, including CIFAR-10 Krizhevsky et al. (2009), ImageNet Deng et al. (2009), and CelebA Liu et al. (2015). CIFAR-10 consists of 60,000 color images of size 32×32 across 10 object classes, representing general-purpose natural scenes. ImageNet is a large-scale visual database with over 14 million human-annotated images spanning

Table 3: Robust Accuracy (%) results on CelebA under PGD-10. The best results are **bolded**, and the second-best results are underlined.

| Method | AF | FN | MFN |
|---|---|---|---|
| w/o Defense | 0.0 | 0.3 | 0.0 |
| GaussianBlur ($\sigma = 7.0$) | 2.8 | 51.4 | 2.8 |
| SHIELD Das et al. (2018) | 17.3 | 84.1 | 27.6 |
| OSCP Lei et al. (2025) | **86.8** | 97.8 | 84.9 |
| DBLP (Ours) | 82.4 | **98.8** | **91.0** |

more than 20,000 categories. CelebA contains over 200,000 celebrity face images, each annotated with 40 facial attributes and five landmark points.

**Training Settings** For our pretrained diffusion backbone, we use Stable Diffusion v1.5 Rombach et al. (2022). The distillation process is trained for 20,000 iterations with a batch size of 4, a learning rate of 8e-6, and a 500-step warm-up schedule. For our leapfrog solver $\Psi_{\text{leap}}$, we set $k = 20$ in Equation equation 13 and $h = 0.8$ in Equation equation 15. During training, adversarial noise is generated using PGD-100 with $\epsilon = 4/255$, targeting a ResNet-50 He et al. (2016) classifier.

**Evaluation Metrics** We evaluate our approach using multiple metrics: clean accuracy (performance on clean data), robust accuracy (performance under adversarial attack), inference time, and image quality metrics including LPIPS Zhang et al. (2018), PSNR, and SSIM Horé & Ziou (2010).

## 5.2 RESULTS

**CIFAR-10** We first conduct experiments on the CIFAR-10 dataset, evaluating our method under adversarial threats constrained by $\ell_\infty$, $\ell_1$, and $\ell_2$ norms. Since DBLP is trained under $\ell_\infty$ attacks, this scenario is considered a seen threat, while the $\ell_1$ and $\ell_2$ settings are treated as unseen threats. The

Table 4: Robust Accuracy (%) on DBLP under Diff-PGD-10 attack $\epsilon = 8/255$ on ImageNet.

| Method | ResNet-50 | ResNet-152 | WideResNet-50-2 | ConvNeXt-B | ViT-B-16 | Swin-B |
|---|---|---|---|---|---|---|
| DiffPure Nie et al. (2022) | 53.8 | 49.4 | 52.2 | 42.9 | 16.6 | 45.1 |
| OSCP Lei et al. (2025) | 59.0 | 56.5 | 57.9 | 49.1 | 34.1 | 53.9 |
| DBLP (Ours) | **63.0** | **59.4** | **60.7** | **52.4** | **38.2** | **58.3** |

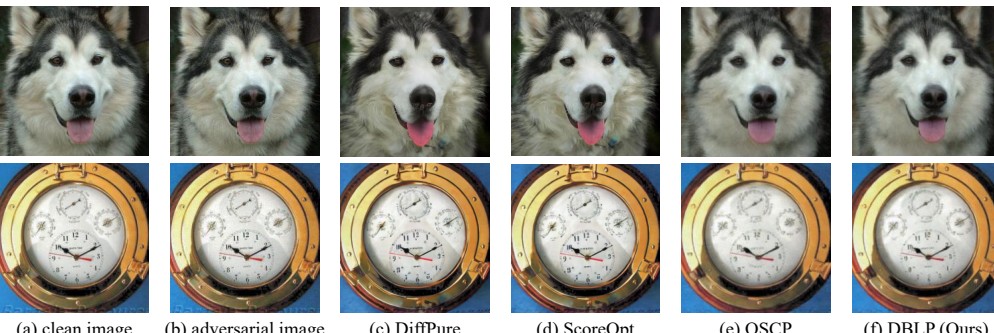

|     |     |     |     |     |     |
|:---:|:---:|:---:|:---:|:---:|:---:|
| (a) clean image | (b) adversarial image | (c) DiffPure | (d) ScoreOpt | (e) OSCP | (f) DBLP (Ours) |

Figure 2: Visualization of (a) clean images, (b) adversarial images and (c-f) purified images under different method.

results are presented in Table 1. Although DBLP belongs to the category of adversarial purification methods, its access to the victim classifier makes it comparable to SOTA adversarial training and purification methods. Adversarial training performs well on seen threats but generalizes poorly to unseen ones, and DiffPure variants offer limited gains. In contrast, DBLP achieves substantially higher robust accuracy on both seen and unseen threats, while preserving strong clean accuracy. It outperforms prior methods by 7.23%, highlighting its robustness, generalization, and efficiency.

**ImageNet** We further conducted comprehensive experiments on the ImageNet dataset, with results summarized in Table 2. Compared to CIFAR-10, adversarial purification methods on this larger-scale dataset can achieve standard accuracy comparable to or even surpassing that of adversarial training, while offering substantially higher robust accuracy. Notably, our method, DBLP, consistently achieves strong performance across various adversarial attacks. Under PGD-100, AutoAttack, and PGD-200 (with $\epsilon = 8/255$), DBLP outperforms previous SOTA approaches by 1.14%, 0.64%, and 0.04% on average, respectively, in terms of both standard and robust accuracy. These results demonstrate the scalability, robustness, and general applicability of DBLP across datasets of different complexity and size.

**Celeb-A** We further validated the effectiveness of our method on a subset of the CelebA-HQ dataset by evaluating it against three representative victim models: ArcFace (AF) Deng et al. (2019), FaceNet (FN) Schroff et al. (2015), and MobileFaceNet (MFN) Chen et al. (2018). Leveraging model weights pretrained on ImageNet, we applied our purification framework to adversarial face images. As shown in Table 3, DBLP significantly enhances purification performance on facial data, demonstrating its robust generalization across image resolutions and domains.

### 5.3 TRANSFERABILITY

We further evaluated DBLP under the Diff-PGD attack Xue et al. (2023). The LCM was trained using PGD-generated adversarial noise on ResNet-50 and tested for transfer robustness across diverse architectures, including ResNet-50/152, WideResNet-50-2 Zagoruyko & Komodakis (2016), ConvNeXt-B Liu et al. (2022), ViT-B-16 Kolesnikov et al. (2021), and Swin-B Liu et al. (2021). As shown in Table 4, DBLP consistently outperforms prior SOTA methods under Diff-PGD-10, demonstrating strong cross-architecture robustness.

Table 5: Inference time of purification models to purify one image. The best results are **bolded**.

| Method | runtime (s) |
|---|---|
| GDMP Wang et al. (2022) | $\sim 43$ |
| DiffPure Nie et al. (2022) | $\sim 53$ |
| OSCP Lei et al. (2025) | $\sim 0.8$ |
| DBLP (Ours) | $\sim \mathbf{0.2}$ |

Table 6: Ablation study of adaptive semantic enhancement.

| | Robust Acc.↑ | LPIPS↓ | SSIM↑ |
|---|---|---|---|
| w/o Edge Map | 74.2 | 0.1386 | 0.7409 |
| Edge Map | 74.8 | 0.1172 | 0.7430 |
| DBLP (Ours) | **75.6** | **0.1012** | **0.7655** |

## 5.4 Inference Time

A key limitation of diffusion-based adversarial purification is the long inference time, which impedes real-time deployment. As shown in Table 5, DBLP achieves SOTA inference speed and significantly outperforms other methods. On ImageNet, it completes purification in just 0.2 seconds, greatly accelerating diffusion-based defenses and enabling practical real-time use.

## 5.5 Image Quality

Beyond correct classification, adversarial purification also seeks to maintain visual fidelity relative to the clean input. As shown in Table 7, DBLP achieves strong image quality across all three metrics, with purified outputs $\mathbf{x}_{pur}$ closely matching both adversarial $\mathbf{x}_{adv}$ and clean images $\mathbf{x}$. This highlights DBLP's superior visual quality. Qualitative results in Figure 2 further confirm its ability to preserve fine-grained details.

Table 7: A quality comparison between the clean image $\mathbf{x}$ and the purified image $\mathbf{x}$pur. The $\mathbf{x}_{adv}$ row reports the metrics between the purified and adversarial images. The best results are **bolded**.

| Method | LPIPS↓ | PSNR↑ | SSIM↑ |
|---|---|---|---|
| $\mathbf{x}_{adv}$ | 0.0975 | 26.17 | 0.7764 |
| DiffPure Nie et al. (2022) | 0.2616 | 24.11 | 0.7155 |
| OSCP Lei et al. (2025) | 0.2370 | 24.13 | 0.7343 |
| DBLP (Ours) | **0.1012** | **26.03** | **0.7655** |

## 5.6 Ablation Study

We conduct ablation studies to assess the adaptive semantic enhancement module in DBLP, with results in Table 6. Omitting edge maps leads to a small drop in robust accuracy but a significant decline in image quality. Using pyramid edge maps further improves both metrics, showing that multi-scale edge representations better capture structural details and enhance visual fidelity. We further conduct a parameter analysis on the number of inference steps, as shown in Figure 3. As the number of steps increases, robust accuracy shows a slight improvement, while sampling time grows significantly. For more ablation results, please refer to Appendix A.3.

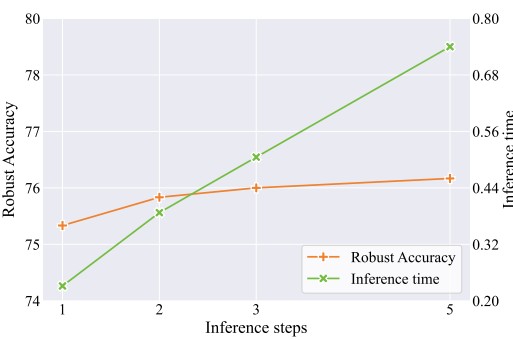

Figure 3: Parameter analysis of inference steps.

## 6 Conclusion

In this work, we propose DBLP, an efficient diffusion-based adversarial purification framework. By introducing noise bridge distillation into the LCM, DBLP establishes a direct bridge between the adversarial and clean data distributions, significantly improving both robust accuracy and inference efficiency. Additionally, the adaptive semantic enhancement module fuses pyramid edge maps as conditional for LCM, leading to superior visual quality in purified images. Together, these advancements bring the scientific community closer to practical, real-time purification systems.

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

## A DERIVATIONS AND PROOFS

### A.1 PROOF OF LIMIT FORMULA

Here, we aim to show that as $t \to 0$, the difference between the consistency model outputs converges to the adversarial perturbation, i.e., $\boldsymbol{f_\theta}(\mathbf{z}_t^a, \varnothing, t) - \boldsymbol{f_\theta}(\mathbf{z}_t, \varnothing, t) \to \boldsymbol{\epsilon}_a$.

$$
\lim_{t \to 0} \boldsymbol{f_\theta}(\mathbf{z}_t^a, \varnothing, t) - \boldsymbol{f_\theta}(\mathbf{z}_t, \varnothing, t)
$$

$$
= \lim_{t \to 0} c_{\text{skip}}(t)\mathbf{z}_t^a + c_{\text{out}}(t)\left(\frac{\mathbf{z}_t^a - \sigma_t \hat{\epsilon}_\theta\left(\mathbf{z}_t^a, c, t\right)}{\alpha_t}\right)
$$

$$
- c_{\text{skip}}(t)\mathbf{z}_t - c_{\text{out}}(t)\left(\frac{\mathbf{z}_t - \sigma_t \hat{\epsilon}_\theta\left(\mathbf{z}_t, c, t\right)}{\alpha_t}\right)
$$

$$
= \lim_{t \to 0} c_{\text{skip}}(t)\left(\mathbf{z}_t^a - \mathbf{z}_t\right)
$$

$$
+ c_{\text{out}}(t)\left(\frac{\left(\mathbf{z}_t^a - \mathbf{z}_t\right) - \sigma_t(\hat{\epsilon}_\theta\left(\mathbf{z}_t^a, c, t\right) - \hat{\epsilon}_\theta\left(\mathbf{z}_t, c, t\right))}{\alpha_t}\right)
$$

$$
= \lim_{t \to 0} c_{\text{skip}}(t)\sqrt{\bar{\alpha}_t}\boldsymbol{\epsilon}_a + c_{\text{out}}(t)\left(\frac{\sqrt{\bar{\alpha}_t}\boldsymbol{\epsilon}_a - \sigma_t(\hat{\epsilon}_\theta^a - \hat{\epsilon}_\theta)}{\alpha_t}\right)
$$

$$
= \lim_{t \to 0} c_{\text{skip}}(t)\sqrt{\bar{\alpha}_t}\boldsymbol{\epsilon}_a
$$

$$
= \boldsymbol{\epsilon}_a
$$

(19)

### A.2 DERIVATION OF EQUATION EQUATION 10

In Equation equation 9, our objective is to select kt such that the adversarial perturbation $\boldsymbol{\epsilon}_a$ is effectively removed, given that only the adversarial latent $\mathbf{z}^a$ is available at inference. Following Dhariwal & Nichol (2021), we leverage Bayes' theorem and the properties of Gaussian distributions to rewrite the sampling formulation as:

$$
q\left(\tilde{\mathbf{z}}_{t-1}|\tilde{\mathbf{z}}_t, \mathbf{z}_0\right) = \frac{q\left(\mathbf{z}_0, \tilde{\mathbf{z}}_{t-1}, \tilde{\mathbf{z}}_t\right)}{q\left(\mathbf{z}_0, \tilde{\mathbf{z}}_t\right)}
$$

$$
= q\left(\tilde{\mathbf{z}}_t|\tilde{\mathbf{z}}_{t-1}, \mathbf{z}_0\right)\frac{q\left(\tilde{\mathbf{z}}_{t-1}|\mathbf{z}_0\right)}{q\left(\tilde{\mathbf{z}}_t|\mathbf{z}_0\right)}
$$

$$
\propto \exp\left(-\frac{1}{2}\left(\text{A}\left(\tilde{\mathbf{z}}_{t-1}\right)^2 + \text{B}\tilde{\mathbf{z}}_{t-1} + \text{C}\right)\right)
$$

(20)

where,

$$
\text{A} = \frac{1 - \bar{\alpha}_t}{(1 - \alpha_t)(1 - \bar{\alpha}_{t-1})}
$$

$$
\text{B} = -2\sqrt{\alpha_t} \cdot \frac{\tilde{\mathbf{z}}_t - (\sqrt{\alpha_t}k_{t-1} - k_t)\boldsymbol{\epsilon}_a}{1 - \alpha_t}
$$

$$
- 2\frac{\sqrt{\bar{\alpha}_{t-1}}\mathbf{z}_0 + (\sqrt{\bar{\alpha}_{t-1}} - k_{t-1})\boldsymbol{\epsilon}_a}{1 - \bar{\alpha}_{t-1}}
$$

(21)

To achieve our goal, we should remove terms related to $\boldsymbol{\epsilon}_a$ in Equation equation 20, which leads to:

$$
\frac{\sqrt{\alpha_t}(\sqrt{\alpha_t}k_{t-1} - k_t)\boldsymbol{\epsilon}_a}{1 - \alpha_t} = \frac{(\sqrt{\bar{\alpha}_{t-1}} - k_{t-1})\boldsymbol{\epsilon}_a}{1 - \bar{\alpha}_{t-1}}
$$

(22)

Then we have:

$$
k_t = \frac{\bar{\alpha}_t - 1}{\sqrt{\alpha_t}(\bar{\alpha}_{t-1} - 1)}k_{t-1} + \frac{\sqrt{\bar{\alpha}_{t-1}}(1 - \alpha_t)}{\sqrt{\alpha_t}(\bar{\alpha}_{t-1} - 1)}
$$

$$
= \frac{\sqrt{\alpha_t}(\bar{\alpha}_t - 1)}{\bar{\alpha}_t - \alpha_t}k_{t-1} + \frac{\sqrt{\bar{\alpha}_{t-1}}(1 - \alpha_t)}{\bar{\alpha}_t - \alpha_t}
$$

(23)

By dividing both sides by $\sqrt{\bar{\alpha}_t}$, we obtain the recursive formula:

$$\frac{k_t}{\sqrt{\bar{\alpha}_t}} = \frac{\bar{\alpha}_t - 1}{\bar{\alpha}_t - \alpha_t} \frac{k_{t-1}}{\sqrt{\bar{\alpha}_{t-1}}} + \frac{1 - \alpha_t}{\bar{\alpha}_t - \alpha_t} \tag{24}$$

Thus, we can easily obtain the closed-form expression:

$$\frac{k_t}{\sqrt{\bar{\alpha}_t}} = \frac{\bar{\alpha}_1(1 - \bar{\alpha}_t)}{\bar{\alpha}_t(1 - \bar{\alpha}_1)} \left( \frac{k_1}{\sqrt{\bar{\alpha}_1}} - 1 \right) + 1 \tag{25}$$

With $k_T = 0$, replace $t = T$ in the equation, we have:

$$\frac{k_1}{\sqrt{\bar{\alpha}_1}} = 1 - \frac{\bar{\alpha}_T(1 - \bar{\alpha}_1)}{\bar{\alpha}_1(1 - \bar{\alpha}_T)} \tag{26}$$

Finally we can obtain the closed-form expression of $k_t$:

$$k_t = \sqrt{\bar{\alpha}_t} \left( 1 - \frac{\bar{\alpha}_T(1 - \bar{\alpha}_t)}{\bar{\alpha}_t(1 - \bar{\alpha}_T)} \right) = \sqrt{\bar{\alpha}_t} - \frac{\bar{\alpha}_T(1 - \bar{\alpha}_t)}{\sqrt{\bar{\alpha}_t}(1 - \bar{\alpha}_T)} \tag{27}$$

### A.3 MORE ABLATION RESULTS

We conducted additional ablation studies to rigorously evaluate the effectiveness of each component in DBLP. Specifically, we ablated Noise Bridge Distillation (NBD) and the Leapfrog ODE solver, comparing them respectively with the consistency distillation loss (CD) of the Latent Consistency Model and the conventional DDIM solver. The distillation loss ablation results, summarized in Table 8, demonstrate that NBD consistently outperforms the traditional CD across all metrics, achieving superior robust accuracy and perceptual image quality. This indicates that, by introducing a noise bridge, NBD more effectively aligns the adversarial noise distribution with the clean data distribution, thereby substantially enhancing both model robustness and the quality of purified images.

Table 8: Ablation study on different distillation loss.

| Distillation Loss | Robust Accuracy | LPIPS | SSIM |
|---|---|---|---|
| w/o | 65.1 | 0.1857 | 0.7260 |
| CD | 73.5 | 0.1337 | 0.7492 |
| NBD | 75.6 | 0.1012 | 0.7655 |

In the ODE solver ablation, the Leapfrog solver exhibits remarkable performance and efficiency, surpassing the DDIM solver. These results confirm that the Leapfrog solver's distinctive update mechanism enables higher computational efficiency without compromising purification quality.

Table 9: Ablation study on different ODE solvers.

| ODE Solver | Robust Accuracy | LPIPS | Time |
|---|---|---|---|
| DDIM | 75.40 | 0.1029 | 0.2670 |
| Leapfrog | 75.60 | 0.1012 | 0.2315 |

