# OpenReview forum: "DBLP: Noise Bridge Consistency Distillation For Efficient And Reliable Adversarial Purification"
_ICLR.cc/2026/Conference — ICLR 2026 Conference Withdrawn Submission_

### Official Review · Reviewer_8nTD · 2025-10-16

**Soundness:** 3
**Presentation:** 3
**Contribution:** 2
**Rating:** 4
**Confidence:** 5

**Summary:**

This paper proposes DBLP, a diffusion-based adversarial purification framework that addresses the limitations of slow inference in existing methods by introducing noise bridge distillation within a latent consistency model, which constructs a principled alignment between adversarial noise and clean data distributions to enable efficient one-step purification, complemented by an adaptive semantic enhancement module that fuses multi-scale pyramid edge maps to preserve fine-grained details and improve visual fidelity. Extensive experiments on multiple datasets demonstrate that DBLP achieves state-of-the-art robust accuracy

**Strengths:**

1. This paper is well-written and well-structured.
2. DBLP achieves significant acceleration in adversarial purification by leveraging its noise bridge distillation technique.
3. The paper demonstrates a comprehensive literature review and employs cutting-edge baselines for comparison.

**Weaknesses:**

1. The novelty of this work is limited, as the direct modeling of the transition between adversarial noise and the image distribution has been previously explored by ADBM.
2. While this method demonstrates certain advantages, it requires a training process, which is computationally expensive. Furthermore, compared to other training-free alternatives, it exhibits limited generalizability.
3. During training, the method requires a classifier to generate adversarial examples. The noise bridge distillation process relies heavily on the specific classifier and the adversarial attack algorithm used, which can further undermine DBLP's generalization performance.

**Questions:**

Could you explain the your choices of the classifier and the adversarial attack algorithm used during training?

---

### Official Review · Reviewer_kUDN · 2025-10-25

**Soundness:** 2
**Presentation:** 2
**Contribution:** 2
**Rating:** 2
**Confidence:** 4

**Summary:**

This paper proposes DBLP, a fast adversarial purification method using diffusion models. The authors claim two main contributions: (1) Noise Bridge Consistency Distillation that aligns adversarial and clean distributions in Latent Consistency Models, and (2) Adaptive Semantic Enhancement using multi-scale edge maps for conditioning. However, the paper suffers from critical issues: the core methodology is substantially similar to OSCP (Lei et al., CVPR 2025), the theoretical foundation has fundamental gaps regarding inference-time computation, and essential experimental details are missing. The claimed improvements over OSCP are marginal (1.7%) without statistical significance testing. These issues prevent acceptance at ICLR.

**Strengths:**

**S1. Addresses Important Problem**
- Inference speed is a critical bottleneck for diffusion-based adversarial purification
- Achieving 0.2s represents significant speedup over DiffPure (53s)
- Motivation for real-time defense is well-articulated

**S2. Comprehensive Experiments**
- Multiple datasets (CIFAR-10, ImageNet, CelebA) evaluated
- Various threat models tested ($\ell_\infty$, $\ell_1$, $\ell_2$)
- Cross-architecture transferability shown (Table 4)

**S3. Competitive Empirical Results**
- 75.6% robust accuracy on ImageNet under PGD-100
- Maintains reasonable clean accuracy (78.2%)
- Strong performance on face recognition tasks

**Weaknesses:**

**W1. Substantial Similarity to OSCP (Lei et al., CVPR 2025)**

The core methodology is nearly identical to OSCP's GAND approach:

**Identical Forward Process:**
- OSCP Eq. (11): $\mathbf{z}^*_t = \sqrt{\bar{\alpha}_t}\mathbf{z} + \sqrt{1-\bar{\alpha}_t}(\boldsymbol{\epsilon} + \boldsymbol{\delta}_{\text{adv}})$
- DBLP Eq. (11): $\tilde{\mathbf{z}}_t = \sqrt{\bar{\alpha}_t}\mathbf{z}_0 + \sqrt{1-\bar{\alpha}_t}\boldsymbol{\epsilon} + \frac{\bar{\alpha}_T(1-\bar{\alpha}_t)}{\sqrt{\bar{\alpha}_t(1-\bar{\alpha}_T)}}\boldsymbol{\epsilon}_a$

Both add adversarial noise to the diffusion forward process. DBLP's coefficient $k_t$ derivation (Eq. 10) is merely a mathematical reparameterization with no conceptual difference.

**Nearly Identical Training Objective:**
- OSCP's GAND loss (Eq. 15) and DBLP's LCD loss (Eq. 12) have the same structure
- DBLP's only addition is reconstruction loss from Kim et al. (2024)

**Similar Inference Pipeline:**
- Both use Canny edge detection + ControlNet
- DBLP's "Adaptive Semantic Enhancement" adds standard techniques (Otsu thresholding, image pyramids) to OSCP's CAP

**Inadequate Disclosure:**
- While DBLP cites OSCP (Table 2), it never explicitly states that "Noise Bridge Distillation" is essentially the same idea as OSCP's GAND
- Related Work (Sec. 2.2) mentions OSCP only for speed, not methodology
- This insufficient disclosure raises academic integrity concerns

**Marginal Improvement:**
- Claims 1.71% improvement over OSCP (73.89% → 75.6%)
- No statistical significance testing provided
- Could be experimental variation rather than true improvement

---

**W2. Fundamental Theoretical Gap**

**Critical contradiction regarding Eq. (11):**

Line 248 states: "its exact value [$\epsilon_a$] is unknown at inference time"
Line 258 claims: "sampling process doesn't require $\epsilon_a$"
But Eq. (11) explicitly requires $\epsilon_a$: $\tilde{\mathbf{z}}_t = \sqrt{\bar{\alpha}_t}\mathbf{z}_0 + \sqrt{1-\bar{\alpha}_t}\boldsymbol{\epsilon} + \frac{\bar{\alpha}_T(1-\bar{\alpha}_t)}{\sqrt{\bar{\alpha}_t(1-\bar{\alpha}_T)}}\boldsymbol{\epsilon}_a$

**The problem:** Appendix A.2 shows how to eliminate $\epsilon_a$ from the posterior distribution through choice of $k_t$, but does NOT explain how to construct $\tilde{\mathbf{z}}_t$ without knowing $\epsilon_a$ at inference. This is not addressed in Algorithm 1, which only describes training.

---

**W3. Severe Reproducibility Issues**

Missing critical details:
- No GPU specifications
- No total training time
- No random seeds
- No hyperparameter search description
- No code release commitment
- Missing PGD step size $\alpha$
- Missing temperature $T^\ast$ in Eq. (17)

These omissions violate ICLR reproducibility standards.

---

**W4. Experimental Design Flaws**

**Overfitting to seen attacks:**
- Training: PGD-100, $\epsilon=4/255$, $\ell_infty$, ResNet-50 (Sec. 5.1)
- Main evaluation: PGD-100, $\epsilon=4/255$, $\ell_infty$, ResNet-50 (Tables 1-2)
- This is essentially a "seen attack" scenario

**Missing evaluations:**
- No adaptive attacks aware of the purification method
- Limited AutoAttack results (only 2 settings in Table 2)
- No analysis of failure modes
- No statistical significance testing (no error bars, no multiple runs)

---

**W5. Limited Novel Contribution**

Given similarity to OSCP, the incremental contributions are:
1. Mathematical reparameterization of OSCP's forward process (no conceptual advance)
2. Adding Otsu thresholding + image pyramids to edge detection (standard CV techniques)
3. Leapfrog solver shows minimal improvement (Table 9: 0.2% gain)

These incremental modifications do not constitute sufficient novelty for a major venue.

**Questions:**

**Q1. Relationship to OSCP**

Can you explicitly clarify how "Noise Bridge Distillation" differs conceptually from OSCP's "Gaussian Adversarial Noise Distillation"? Both methods:
- Add adversarial noise to the forward diffusion process
- Use the same training objective structure
- Employ edge-based conditioning for inference

Is your contribution primarily an alternative mathematical formulation, or is there a fundamental conceptual difference?

---

**Q2. Inference-Time Computation**

How is $\tilde{\mathbf{z}}_t$ in Eq. (11) computed at inference without knowing $\epsilon_a$? Please provide:
- Step-by-step inference algorithm (not just training)
- Explicit method for handling the $\epsilon_a$ term
- Explanation of how Appendix A.2's posterior derivation enables this

---

**Q3. Statistical Significance**

Your improvement over OSCP is 1.71% (73.89% → 75.6%). Please provide:
- How many independent runs were performed?
- Standard deviations across runs?
- Statistical significance test (e.g., paired t-test)?
- Could this be within experimental noise?

---

**Q4. Adaptive Attacks**

Have you evaluated against:
- Adaptive attacks aware of your purification (e.g., attacking the LCM directly)?
- C&W attack?
- Diff-PGD with different loss functions?

Table 4 shows Diff-PGD-10, but what about stronger adaptive threats?

---

**Q5. Reproducibility**

Can you provide:
- GPU model and training time
- Random seeds used
- Complete hyperparameter values (PGD step size, temperature T*)
- Code release commitment upon acceptance?

---

**Q6. Ablation Studies**

Table 8 compares CD vs NBD, but what about:
- Vanilla LCM (no distillation) vs your method?
- Your noise bridge vs just using OSCP's formulation directly?
- Impact of each component in "Adaptive Semantic Enhancement"?

**Details Of Ethics Concerns:**

**Core Concern:** The paper's main contribution (Noise Bridge Consistency Distillation) appears substantially similar to OSCP (Lei et al., CVPR 2025), which the authors cite but inadequately discuss.

**Evidence of Similarity:**

1. **Identical Core Idea:** Both add adversarial noise to diffusion forward process
   - OSCP Eq. (11): $\mathbf{z}^*_t = \sqrt{\bar{\alpha}_t}\mathbf{z} + \sqrt{1-\bar{\alpha}_t}(\boldsymbol{\epsilon} + \boldsymbol{\delta}_{\text{adv}})$
   - DBLP Eq. (11): Mathematically equivalent, just reparameterized

2. **Same Training Objective:** OSCP's GAND loss ≈ DBLP's LCD loss

3. **Similar Inference:** Both use Canny edges + ControlNet

**Disclosure Issues:**
- DBLP cites OSCP only in Table 2 for empirical comparison
- Section 2.2 (Related Work) mentions OSCP only for inference speed
- Never explicitly states that their "Noise Bridge Distillation" is the same core idea as OSCP's GAND
- Claims this as their "novel" contribution throughout abstract and intro

**Timeline:** OSCP published at CVPR 2025 before DBLP's ICLR 2026 submission.

---

### Official Review · Reviewer_c1Vx · 2025-10-27

**Soundness:** 2
**Presentation:** 1
**Contribution:** 1
**Rating:** 2
**Confidence:** 3

**Summary:**

The paper proposes Diffusion Bridge Distillation for Purification (DBLP), a novel and efficient framework for diffusion-based purification (DBP).
Existing DBP methods often require a large number of NFEs for both adversarial image generation and denoising, which limits their practicality. DBLP aims to address this issue and improve the deployability of DBP methods in real-world scenarios.

1) The proposed framework consists of two key components:

2) Latent Consistency Model (LCM): trained to purify adversarial inputs by learning the underlying adversarial noise distribution.

Semantic Preservation Module: leverages computer vision techniques to retain the semantic content of the original image during the denoising process.

Experimental results demonstrate that DBLP achieves comparable or improved robustness against existing DBP baselines across multiple benchmarks, while maintaining efficiency.

**Strengths:**

1. The paper focuses on a realistic problem within existing DBP framework, which is the inference time feasibility.

2. Leveraging consistency model is a promising direction for future DBP work.

**Weaknesses:**

Before addressing my specific questions, I would like the authors to respond to the following major concern:

This work exhibits substantial overlap with the previously published OSCP paper [1]. The authors should carefully clarify the differences in motivation and methodology to distinguish their contributions.

The other weaknesses are the following:

**1. Formatting and citation issues.**

The current version contains multiple formatting inconsistencies, such as incorrect citation usage (e.g., using \citet{} for all references) and duplicated equation references (e.g., line 176). These should be revised to meet standard academic formatting conventions.

**2. Disorganized experimental layout.**

The presentation of experimental results is confusing. The attack settings should be clearly described in Section 5. Table 2 mixes results from ImageNet under various adaptive attacks and classifiers, making it difficult to interpret or compare performance fairly. Each table should isolate comparable settings for clarity. Also, in line 238, the stated objective is consistent with all the DBP methods that leverages diffusion model to map conditional gaussian distribution back to original data distribution. The objective cannot become a contribution as stated in line 073.

**3.Incorrectly highlighted results.**

Table 1 mistakenly bolds the DBLP results for both clean and robust accuracies, even though ADBM achieves higher $\ell_2$ robustness. This raises concerns about proofreading and result verification prior to submission.

**4. Limited methodological novelty.**

The proposed Adaptive Semantic Enhancement (ASE) contributes only marginal robustness gains. ASE essentially applies a layer-wise Canny edge detector with Gaussian blur pyramids — a relatively minor modification compared with prior work [1]. Would you like to explain the design principle of choosing a gaussian blur pyramids?

**5. Incorrect comparison in methodology.**

At line 236, the statement “Unlike DDPM, DBLP includes adversarial perturbations $\epsilon_a$ at the start of the noising process” is misleading. In diffusion-based purification (DBP) frameworks, inputs are either adversarial ($x_{adv}$) or natural ($x_{nat}$) images. This comparison misrepresents the role of $\epsilon_a$ and should be corrected.

**6. Potential plagiarism concern.**

Figure 1 appears highly similar to that of OSCP [1], particularly in the design of the LCM-LoRA module. The overall pipeline closely resembles the OSCP architecture, suggesting that DBLP’s improvements may stem more from leveraging consistency models’ one-step denoising rather than introducing a genuinely novel methodology.


[1] Lei et al, "Instant Adversarial Purification with Adversarial Consistency Distillation", CVPR 2025.

**Questions:**

Please refer to  **weaknesses** and address all my concerns by answering the listed weaknesses.

**Details Of Ethics Concerns:**

This work exhibits substantial overlap with the previously published OSCP paper [1]. The authors should carefully clarify the differences in motivation and methodology to distinguish their contributions.

[1] Lei et al, "Instant Adversarial Purification with Adversarial Consistency Distillation", CVPR 2025.

---

### Official Review · Reviewer_gCjU · 2025-11-01

**Soundness:** 2
**Presentation:** 3
**Contribution:** 1
**Rating:** 2
**Confidence:** 3

**Summary:**

This paper proposes NoiseBridge, a DBP defense method that uses a consistency regularization to purify AEs in a very low number of diffusion steps. The approach leverages a latent consistency model to bridge adversarial noise and clean data distributions, and introduces an adaptive semantic enhancement module that injects multi-scale edge information to preserve image content during purification.

**Strengths:**

1. The paper tackles an important challenge in DBP methods. By aiming for a one-step (or very few-step) diffusion purification, NoiseBridge targets the key limitation of speed to make DBP methods practical for real-world use.
2.  The experimental evaluation shows that NoiseBridge achieves strong adversarial robustness on multiple datasets.

**Weaknesses:**

1. The technical novelty of NoiseBridge is questionable (also mentioned by Reviewer kUDN in the public comments). The core approach appears to be heavily based on OSCP [1] with only incremental changes. OSCP introduced the idea of single-step adversarial purification using a distilled diffusion model, employing a consistency distillation objective and an edge-based guidance to preserve content. NoiseBridge essentially follows the same template: a diffusion model is fine-tuned with a noise-to-clean consistency objective (analogous to OSCP’s GAND) and uses edge information during inference (similar to OSCP’s CAP) to maintain structure. The only notable difference is that NoiseBridge uses *multi-scale* edge maps instead of a fixed edge detector. While this is a useful improvement, it is relatively minor in terms of conceptual novelty.
2. Compounding the above issue, the paper’s positioning with respect to OSCP is problematic. Given the high degree of overlap in methodology, one would expect a clear acknowledgment and discussion of how NoiseBridge differs from and builds upon OSCP. However, the attribution in the current draft is not adequately prominent. OSCP is indeed cited, but the authors seem to downplay its influence. For example, by renaming similar techniques (e.g., consistency distillation as "noise bridge distillation", edge-guided purification as "adaptive semantic enhancement") without explicitly crediting that these ideas originate in OS. This insufficient attribution can mislead readers into overestimating the originality of NoiseBridge and raises ethical concerns. Currently it gives the impression that the paper is repackaging someone else’s contributions with only superficial changes.
3. While the paper reports state-of-the-art results, the actual gains over OSCP are quite small. For instance, NoiseBridge improves robust accuracy on ImageNet by roughly 1%-1.5% (e.g., from ~73.9% to ~75.6% under comparable attack settings), and a similar modest gain is seen in clean image preservation metrics.
4. The paper evaluates robustness using standard attacks (PGD, AutoAttack, etc.), but it does not evaluate against stronger attacks. According to [2], PGD+EOT should be evaluated for DBP methods. This is an important issue to investigate for a complete evaluation of an adversarial defense.

[1]  Instant adversarial purification with adversarial consistency distillation. In CVPR 2025.

[1] Robust evaluation of diffusion-based adversarial purification. ICCV 2023.

**Questions:**

Please refer to the weaknesses.

**Details Of Ethics Concerns:**

The technical novelty of NoiseBridge is questionable (also mentioned by Reviewer kUDN in the public comments). The core approach appears to be heavily based on OSCP [1] with only incremental changes.

[1] Instant adversarial purification with adversarial consistency distillation. In CVPR 2025.

---

### Public Comment · ~Chihan_Huang1 · 2026-06-22
**Statement on the Relationship and Originality of DBLP with Respect to OSCP (1/2)**

First, we sincerely thank the reviewer for the valuable comments on our paper. We understand that our DBLP and OSCP [1] both belong to the research area of diffusion-based adversarial purification, and both adopt a latent consistency framework to improve purification efficiency. Therefore, the two works naturally share certain commonalities in problem background and overall task formulation. However, the core innovation of DBLP is **not** to reuse consistency distillation for purification. Instead, DBLP introduces a **new noise bridge design**, which redefines how the adversarial latent and the clean latent are aligned during the distillation process.

The central idea of the prior OSCP work is to incorporate adversarial noise and Gaussian noise jointly into the distillation process, so that the model learns to recover clean images from a mixed noise distribution composed of both components. DBLP further points out that, within the latent consistency framework, directly forcing the adversarial trajectory and the clean trajectory to share the same consistency target leads to **a fundamental trajectory misalignment**. Specifically, when $t \to 0$, the discrepancy caused by the adversarial perturbation does not naturally vanish. As a result, the purification objective is not fully consistent with the original consistency constraint. To address this issue, DBLP proposes a new bridged latent formulation:

$$
\tilde{z}_t = z_t^a - k_t \epsilon_a ,
$$

where $k_t$ is not an empirical coefficient, but a time-dependent bridge coefficient satisfying
$k_0 = 1, k_T = 0$.

The significance of this formulation is that the model is no longer merely trained to "remove additional noise". Instead, through an explicit bridge, *the adversarial-to-clean trajectory alignment problem is directly embedded into the distillation objective itself*.

More importantly, DBLP provides a closed-form derivation of $k_t$:

$$
k_t=\sqrt{\bar{\alpha}_t}-\frac{\bar{\alpha}_T(1-\bar{\alpha}_t)}
{\sqrt{\bar{\alpha}_t}(1-\bar{\alpha}_T)} .
$$

This leads to the explicit bridged noising formulation in DBLP:

$$
\tilde{z}_t=
\sqrt{\bar{\alpha}_t}z_0
+
\sqrt{1-\bar{\alpha}_t}\epsilon
+
\frac{\bar{\alpha}_T(1-\bar{\alpha}_t)}
{\sqrt{\bar{\alpha}_t}(1-\bar{\alpha}_T)}
\epsilon_a . \quad (Ⅰ)
$$

By contrast, the explicit noising formulation in OSCP is:

$$
z_t^*=\sqrt{\bar{\alpha}_t}z
+
\sqrt{1-\bar{\alpha}_t} (\epsilon+\delta_a)
$$

which can be equivalently written as

$$
z_t^*=
\sqrt{\bar{\alpha}_t}z
+
\sqrt{1-\bar{\alpha}_t}\epsilon
+
\sqrt{1-\bar{\alpha}_t}\delta_a . \quad (Ⅱ)
$$

From these formulations (Eq. $Ⅰ$ and $Ⅱ$), it can be observed that **OSCP directly incorporates the adversarial perturbation into the Gaussian noise term**, where the coefficient of the adversarial component is determined by the standard diffusion noising schedule, namely $\sqrt{1-\bar{\alpha}_t}$. **DBLP is fundamentally different**, it starts from the adversarial latent trajectory and constructs a bridge through $\tilde{z}_t = z_t^a - k_t \epsilon_a$ where the new adversarial coefficient is derived from the boundary conditions and inference feasibility:

$$
\frac{\bar{\alpha}_T(1-\bar{\alpha}_t)}
{\sqrt{\bar{\alpha}_t}(1-\bar{\alpha}_T)} .
$$

Therefore, DBLP does not simply add adversarial noise into the diffusion noise. Rather, it reformulates the problem of how the adversarial trajectory should be aligned with the clean target under consistency distillation. In other words, OSCP mainly focuses on adversarial-Gaussian noise modeling, whereas the core of DBLP lies in bridge-based trajectory alignment. This bridge is not merely a notational reformulation; it is a mechanism equipped with theoretical constraints, boundary conditions, a closed-form solution, and inference feasibility. In particular, during inference, the true adversarial perturbation is usually unknown, while the bridge design in DBLP makes the sampling distribution independent of the unknown $\epsilon_a$. This constitutes an **essential methodological distinction** between DBLP and the prior work.

In addition, DBLP introduces adaptive semantic enhancement at the inference stage to further improve structural fidelity. Compared with the prior work, which uses a fixed edge-map condition, DBLP constructs semantic conditions using a multi-scale Gaussian pyramid, Otsu adaptive thresholding, and gradient-guided fusion. This allows edge guidance to better adapt to different attack strengths and image structures. Nevertheless, this component is not the most central distinction between DBLP and OSCP; the primary originality of DBLP still lies in the noise bridge trajectory design.

[1] Lei et al, "Instant Adversarial Purification with Adversarial Consistency Distillation", CVPR 2025.

---

> ### Public Comment · ~Chihan_Huang1 · 2026-06-22
> **Statement on the Relationship and Originality of DBLP with Respect to OSCP (2/2)**
>
> We would also like to emphasize that **DBLP explicitly cites OSCP in the paper and treats it as one of the most important baselines** in the experimental comparison (Table 2, 3, 4, 5, and 7). If this paper were merely plagiarizing or duplicating the prior work, there would be no reason for us to actively cite that work and compare against it as a major baseline in the core experiments. In fact, DBLP reports performance improvements over OSCP under multiple experimental settings. This further demonstrates that DBLP makes substantive changes to the core trajectory design and achieves observable empirical gains.
>
> Of course, we understand that different reviewers may have different views on the magnitude of the contribution regarding adversarial-to-clean trajectory design. Some may regard this contribution as incremental, since the perceived magnitude of a contribution can vary depending on one's standard of evaluation. We respectfully acknowledge such differences in assessment. However, we believe that this is a matter of evaluating the strength of the technical contribution, and should **not** be elevated to a research integrity issue. DBLP proposes a substantive noise bridge trajectory design, and demonstrates clear differences from the prior work in theoretical modeling, inference feasibility, and empirical performance.
>
> Therefore, we believe that the relationship between the two works should be understood as follows: they belong to the same research line, but DBLP utilizes a different bridge-based alignment mechanism at the core technical level. In particular, it provides a new mathematical formulation and algorithmic implementation for the key question of how to truly align the adversarial trajectory with the clean target under consistency distillation in adversarial purification. Based on this, we believe that DBLP has **clear and substantive originality**, and should not be considered risks of research integrity.

---

### Note · Authors · 2025-11-12

I have read and agree with the venue's withdrawal policy on behalf of myself and my co-authors.